# Where to Begin? On the Impact of Pre-Training and Initialization in Federated Learning

**John Nguyen**    **Jianyu Wang**    **Kshitiz Malik**    **Maziar Sanjabi**    **Michael Rabbat**

Meta AI

{ngjhn,jianyuwang,kmalik2,maziars,mikerabbat}@meta.com

## Abstract

An oft-cited challenge of federated learning is the presence of heterogeneity. *Data heterogeneity* refers to the fact that data from different clients may follow very different distributions. *System heterogeneity* refers to the fact that client devices have different system capabilities. A considerable number of federated optimization methods address this challenge. In the literature, empirical evaluations usually start federated training from random initialization. However, in many practical applications of federated learning, the server has access to proxy data for the training task that can be used to pre-train a model before starting federated training. We empirically study the impact of starting from a pre-trained model in federated learning using four standard federated learning benchmark datasets. Unsurprisingly, starting from a pre-trained model reduces the training time required to reach a target error rate and enables the training of more accurate models (up to 40%) than is possible when starting from random initialization. Surprisingly, we also find that starting federated learning from a pre-trained initialization reduces the effect of both data and system heterogeneity. We recommend that future work proposing and evaluating federated optimization methods evaluate the performance when starting from random and pre-trained initializations. We also believe this study raises several questions for further work on understanding the role of heterogeneity in federated optimization.

## 1 Introduction

*Federated learning* (FL) has emerged as a popular distributed machine learning paradigm for privately training a shared model across many participants while the training data never leaves the participant devices. This paper empirically investigates the impact of model initialization on federated optimization methods. Previous empirical evaluations of FL methods start federated training from a randomly initialized model. Transfer learning from pre-trained models has become common practice in natural language processing [31, 6] and computer vision [12, 7], yielding state-of-the-art results on many tasks and enabling faster model convergence in the centralized setting. Although public proxy data is available at the server in many applications, few prior works studied the impact of starting federated training from a pre-trained model.

In cross-device FL [18], the primary setting considered in this paper, a central server coordinates a large number of client devices (possibly on the order of hundreds of millions). Each device possesses a local dataset, and the data at different devices follow different distributions, leading to the *data heterogeneity* challenge [18]. Moreover, client devices have different system capabilities, leading to *system heterogeneity*. Finally, devices communicate with the server over low-bandwidth communication links making the performance bottleneck.

Workshop on Federated Learning: Recent Advances and New Challenges, in Conjunction with NeurIPS 2022 (FL-NeurIPS'22). This workshop does not have official proceedings and this paper is non-archival.

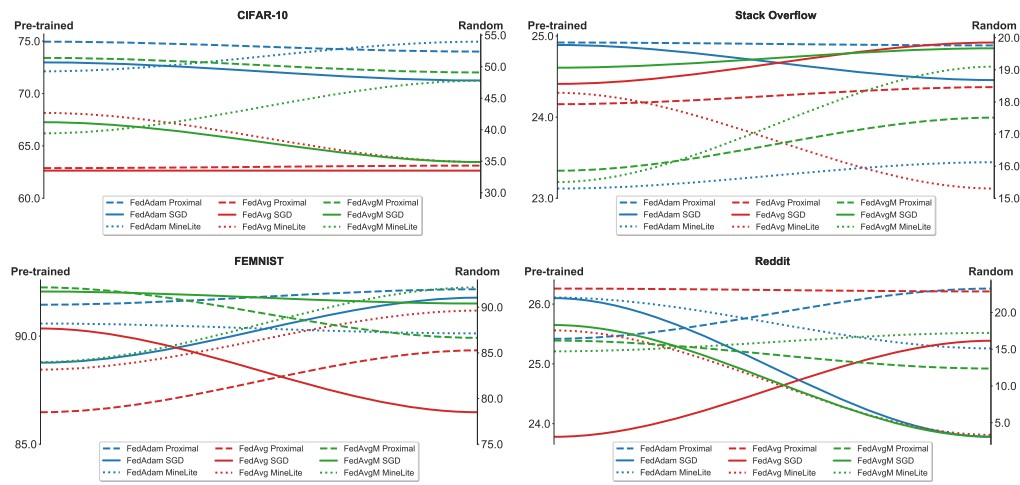

Figure 1: The test set accuracy on four datasets with random and pre-trained weights. We represent SGD by solid lines, PROXIMAL by dashed lines, and MIMELITE by dotted lines. See Section 3.3 and Table 4 in Appendix B for comparison characteristics of each method.

The predominant approach to federated training builds on local update methods such as FEDAVG [26], where a device performs several local updates (e.g., one epoch of SGD on their local training set) before transmitting an update to the server. Although this reduces communication overhead, it can also exacerbate data heterogeneity. Several approaches have been proposed to address this challenge [24, 14, 32, 36, 19, 20, 41, 1]. However, few prior works examine the impact of initialization on federated training.

**Contributions.** In this work, we consider the question: *How does model initialization (random or pre-trained) impact the behavior of federated optimization methods?* We perform an extensive empirical study, comparing 15 variations of federated optimization methods on four commonly-used FL benchmark datasets. Our study reveals three key findings:

1. Although optimizers designed to address heterogeneity typically lead to better performance when starting from a random initialization, when starting from pre-trained model we observe that (cf. Fig. 1): (i) there is not as big a difference between optimizers in terms of accuracy after a fixed number of rounds, and (ii) using an adaptive optimizer at the server, such as FEDADAM, is more important than using any method for addressing heterogeneity.

2. Starting from a pre-trained model significantly reduces the difference between having non-IID vs IID data at clients. Furthermore, when starting from a pre-trained model, the number of local epochs per round can be significantly increased without degrading the final accuracy.

3. The initial loss is not always lower when starting from a pre-trained model. However, the largest Hessian eigenvalue (i.e., local Lipshitz constant, or smoothness) is consistently smaller at initialization when starting from a pre-trained model, compared to when starting from a random initialization.

Some of our empirical observations are consistent with existing FL theoretical convergence guarantees. Our findings also highlight that some aspects of FL are not captured with the existing theory, suggesting directions for future work.

Initializing FL with a pre-trained model can increase final model accuracy and reduce the number of rounds required to achieve a target accuracy. Pre-training leads to communication savings and reduces the overall training time. Figure 2 demonstrates the benefit of pre-training across several datasets (hyperparameters were tuned separately for each dataset–initialization pair; see Section 3 for details of the experimental setup).

Our findings are reproducible using the open-source federated learning framework FLSim [8]. Informed by these findings, we present several recommendations for future research on federated optimization.

---

**Algorithm 1** FedOpt framework

---

1: **Input:** initial global model $x^0$, server and client step sizes $\eta_s$, $\eta_c$, local epochs $E$, rounds $T$
2: **for** each round $t = 1, \ldots, T$ **do**
3:    Server sends $x^{t-1}$ to all clients $i \in \mathcal{S}^t$.
4:    **for** each client $i \in \mathcal{S}^t$ in parallel **do**
5:       Initialize local model $y_i^0 \leftarrow x^{t-1}$.
6:       Each client performs $E$ epochs of local updates via $y_i^{k+1} = \text{CLIENTOPT}(y_i^k, F_i, \eta_c)$. Let $y_i^E$ denote the result after performing $E$ epochs of local updates.
7:       After local training, client $i$ sends $\Delta_i^t = x^{t-1} - y_i^E$ to the server.
8:    **end for**
9:    Server computes aggregate update $\Delta^t = \frac{1}{|\mathcal{S}^t|} \sum_{i \in \mathcal{S}^t} p_i \Delta_i^t$.
10:   Server updates global model $x^t = \text{SERVEROPT}(x^{t-1}, -\Delta^t, \eta_s, t)$.
11: **end for**

---

## 2   Problem Formulation and the FEDOPT framework

We consider the following standard optimization formulation of federated training. We seek to find model parameters $w$ that solve the problem,

$$\min_{w \in \mathbb{R}^d} f(w) := \sum_{i=1}^m p_i F_i(w) \tag{1}$$

where $m$ is the total number of clients, the function $F_i$ measures the average loss of a model with parameters $w$ on the $i$th client's training data, and $p_i > 0$ is the weight given to client $i$. Usually $p_i$ is taken to be proportional to the number of samples at client $i$ so that the optimization problem gives equal weight to all training samples. The goal is to find a model that fits all clients' data well on (weighted) average. In FL, only client $i$ can evaluate $F_i$ and its gradient.

All of the methods we consider in this study can be expressed in the general FEDOPT framework introduced in Reddi et al. [32]; see Algorithm 1. At round $t$, the server sends its last model $x^{t-1}$ to a cohort of clients. Each client in the cohort performs $E$ epochs of local training starting from $x^{t-1}$ using CLIENTOPT with client learning rate $\eta_c$, producing a local model $y_i^E$. Then each client communicates the difference $\Delta_i^t$ between their local model and the server model, where $\Delta_i^t := x^{t-1} - y_i^E$. The server computes a weighted average $\Delta^t$ of the client updates (line 9 in Alg. 1) and updates its own model via $x_{t+1} = \text{SERVEROPT}(x_t, \Delta_t, \eta_s, t)$, where $\text{SERVEROPT}(x_t, \Delta_t, \eta_s, t)$ is a first-order optimizer, $\eta_s$ is the server learning rate, and $t$ is the round number.

## 3   Experimental Setup

### 3.1   Datasets, Models, and Tasks

To facilitate comparing with other work in the literature, we experiment on four standard FL benchmark datasets: CIFAR-10 [22], Federated EMNIST-62 (FEMNIST) [5, 3], Stack Overflow [2] and pushift.io's Reddit[1] [3]. All datasets have a natural non-IID partitioning of the data except for CIFAR-10, for which we use the Dirichlet allocation approach of [14] with parameter 0.1 to partition data across 50 users. For CIFAR-10 and FEMNIST, we train Squeezenet [17] and ResNet18 [11] models, replacing batch norm with group norm [32, 13]. For Stackoverflow and Reddit pushift.io, we use the DistilGPT2 [16] and CharLM [21] models. For additional information about the datasets and models, see Appendix A.1.

### 3.2   Initialization Strategies

We consider two initialization strategies: random initialization and supervised pre-training.

---

[1]A third-party released dataset of Reddit comments from pushshift.io, packaged in the widely-used LEAF benchmark [3]

**Random initialization.** Most prior federated optimization works use random weights to initialize the model. We can use the same random initialization strategies used in the standard (centralized) training of deep networks for each model [17, 16, 11, 21].

**Supervised pre-training.** In many FL applications, pre-training can be done on a large non-private proxy dataset available at the server. To facilitate easily reproducing our results, we use publicly available pre-trained models or pre-train on public data. For tasks using Squeezenet and ResNet18, we use the version of the model pre-trained on ImageNet, available in the PyTorch Torchvision library.[2] For tasks using DistilGPT2, we use the model weights provided in the HuggingFace library that has been distilled from a pre-trained GPT2,[3] and for tasks using CharLM, we pre-train the model on WikiText-103 [28] (see Appendix B.2 for details).

Supervised pre-training is just one possibility, and we leave the investigation of other pre-training strategies (e.g., self-supervised pre-training and meta-learning) as future work; see Section 8.

### 3.3 Algorithms

We compare federated training with five different CLIENTOPT strategies:

**SGD** clients perform standard stochastic gradient descent updates;

**Proximal [24]** clients perform FEDPROX-style local updates; FEDPROX was originally proposed to reduce client drift due to heterogeneity;

**Normalized Averaging [36]** clients use FEDNOVA-style updates and aggregation to compensate for data imbalance across clients;

**MIMELITE [20]** clients make use of an optimizer state (e.g., momentum buffer) from the server during local updates to reduce drift due to data heterogeneity;

**GD** clients perform full-batch gradient updates; in this case, the update $\Delta_i^t$ returned to the server is a full-batch gradient on client $i$'s local training set evaluated at model parameters $x^{t-1}$.

At the server, we consider three strategies for SERVEROPT. In all strategies, the server treats the averaged update $\Delta^t$ as a gradient.

**SGD** the server updates the global model using stochastic gradient descent; when CLIENTOPT is also SGD, this is equivalent to FEDAVG [26].

**SGD with momentum** the server updates the global model using SGD with momentum; when CLIENTOPT is SGD, this is equivalent to FEDAVGM [14].

**Adam** the server updates the global model using the Adam optimizer; when CLIENTOPT is SGD, this is equivalent to FEDADAM [32].

The method commonly referred to as FEDSGD [27] is obtained when CLIENTOPT is full-batch gradient descent (GD) and SERVEROPT is SGD, with $\eta_c = 1$ and $E = 1$.

We focus on the above choices for CLIENTOPT and SERVEROPT because they are reflective of the most widely-cited federated optimization methods, and they also represent a diverse set of possible choices available to the practitioner seeking to deploy cross-device federated training at scale.

### 3.4 Implementation and Tuning

We repeat each experiment with three different seeds and report the average. For each algorithm, model, and dataset, we run a hyperparameter sweep to tune client and server learning rates $\eta_\ell$ and $\eta_g$, and the proximal penalty parameter $\mu$ for FEDPROX; see Appendix A for details. Unless otherwise specified, each client update entails running one local epoch with fixed batch size per task. We perform 1050 rounds of training for Stackoverflow, 1000 rounds of training for CIFAR-10, 1082 training rounds for FEMNIST. See Appendix A.2 for additional details on implementation details. All experiments were performed using the open-source federated learning simulation framework FLSim [8].

---

[2] https://github.com/pytorch/vision
[3] https://huggingface.co/distilgpt2

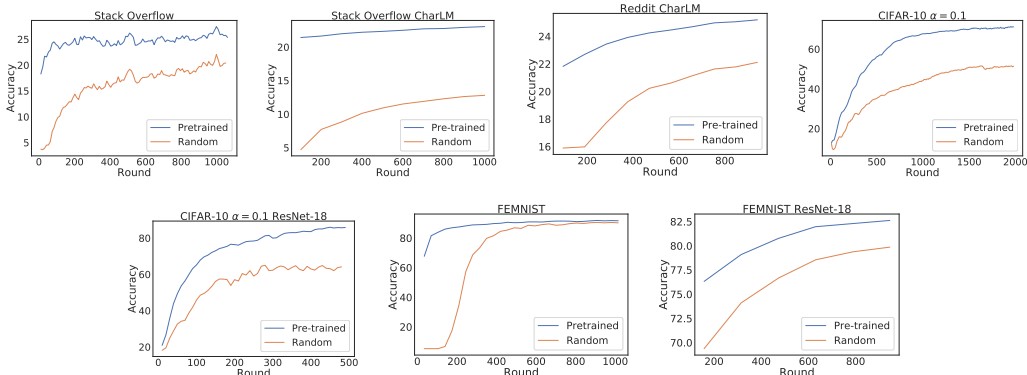

Figure 2: While prior works ignore the importance of initialization, using pre-trained models should be the first step for any practical deployment to save on communication bandwidth and achieve high model accuracy. This figure shows the advantage of using a pre-trained model for four tasks. For Stack Overflow and Reddit, we use DistilGPT2. For CIFAR-10 and FEMNIST, we use SqueezeNet.

## 4  The Impact of Pre-Training in FL

In this section, we illustrate the benefits of pre-training in the federated setting and how pre-training can impact federated optimization algorithms behavior.

**Pre-training changes the ranking of federated optimization algorithms.** If one sorts federated optimization methods based on their performance when starting from a random initialization, the order is substantially different from when using a pre-trained initialization. We focus on nine combinations of SERVEROPT and CLIENTOPT, only using local update methods for CLIENTOPT and excluding full-batch gradient descent. We show the change in performance in Figure 1.

First, observe that the span of final accuracies is much smaller when starting from a pre-trained model. Second, all methods starting with pre-trained models achieve a better accuracy after the same number of steps compared to random models. Lastly, observe that the order of methods changes depending on the initialization. Although no particular method dominates across all workloads in Figure 1, FEDADAM with SGD for CLIENTOPT performs consistently well when starting from a pre-trained model, especially on the two larger language modeling workloads, Stack Overflow and Reddit, and so we focus on studying FEDADAM-SGD below.

**Faster convergence to better accuracy when starting from a pre-trained model.** Figure 2 shows that, as one would hope, when starting from a pre-trained model, it is possible to achieve much better accuracy after a fixed number of steps than when starting federated training from a random initialization. Note that the initial accuracy is not always substantially higher than a random initialization (See Table 2).

**Pre-training closes the accuracy gap between non-IID and IID.** We study how pre-training and data heterogeneity affect convergence without system heterogeneity by fixing the number of local epochs to $E_i = 1$. We compare FEDADAM-SGD under IID and Non-IID data splits. In Figure 4, we report the average accuracy for FedAdam [32] on the four datasets. As expected, randomly initialized models perform much worse than their pre-trained counterparts, and IID partitions yield better quality than non-IID. Surprisingly, the gap between models trained on IID data and models trained on non-IID data is significantly smaller when starting with pre-trained weights. Moreover, pre-training reduces the negative effects of data heterogeneity (i.e., client drift). As a result, we observe that (see Figure 3)

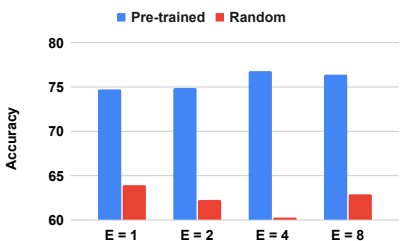

Figure 3: The accuracy for CIFAR-10 using ResNet-18 with increasing number of local epochs.

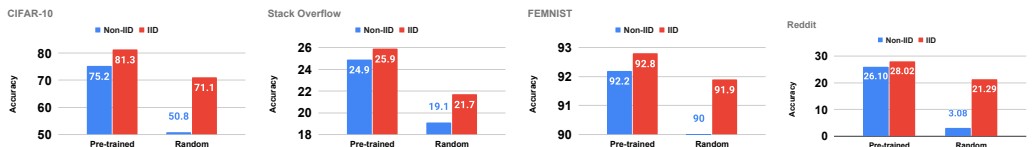

Figure 4: The average accuracy on 3 different seeds for FEDADAM trained on IID and non-IID data. For CIFAR-10 Non-IID, we generate 100 non-IID clients using a Dirichlet(0.1). For other three datasets, we use the natural non-IID client partitions.

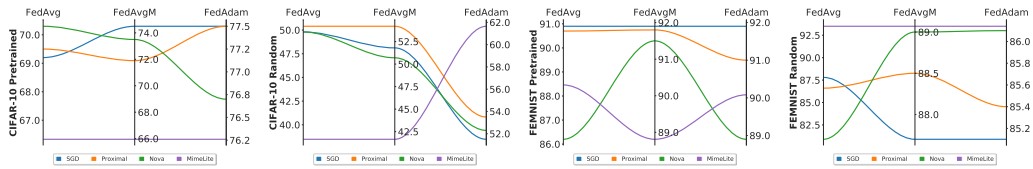

Figure 5: System heterogeneity results comparing FedAvg, FedAvgM and FedAdam with various client optimizers. We simulate system heterogeneity by randomly select 30% of clients per round to perform time-varying local epochs $E_i \sim U(1, 5)$, the same approach as in [36]. FedProx and FedNova correspond to FedAvg with Proximal client optimizer and normalized averaging (NOVA), respectively. We repeat each experiment for 3 different seeds and report the average.

when training from a pre-trained model, increasing the number of local updates does not degrade the final accuracy, in contrast to training from a random model

**FEDADAM GD is as effective as FEDADAM SGD with pre-training.** The seminal work of McMahan et al. [26] shows that taking local SGD steps before server averaging reduces communication by 10-100× compared to taking a full batch gradient step. To understand how pre-training impacts this comparison, we compare FEDADAM with SGD and FEDADAM with GD. While local SGD can reduce communication, the saving is much less when the models are initialized with pre-trained weights compared to random weights. Figure 8 in the Appendix shows that with pre-trained initialization, using GD at the client can yield almost the same result as taking local SGD steps.

**Pre-training reduces the impact of system heterogeneity.** To study the impact of pre-training on system heterogeneity, we follow the setup described in [24, 36]. We sample 30% of clients uniformly at random, and client $i$ performs $E_i$ local epochs where $E_i \sim U(1, 5)$ while the remaining 70% of the clients perform $E_i = 1$ epochs; this models the setting where clients have different processing capabilities and they perform as much work as they can within a given time limit. Figure 5 shows that FEDADAM-SGD consistently outperforms other methods specifically designed for system heterogeneity (NOVA, PROXIMAL, MIMELITE) when starting from a pre-trained model. Apparently using an adaptive optimizer at the server is sufficient to correct for the negative effects of systems heterogeneity when starting from a pre-trained model. On the other hand, when starting from a random initialization, optimizers specifically designed for system heterogeneity (i.e., FEDNOVA) outperform SGD (Figure 5 right). Moreover, the accuracy gap between algorithms is more pronounced in the random initialization setting, whereas in the pre-trained setting, all algorithms converge to more similar accuracies. Our results suggest that pre-training may reduce the need for algorithms that try to correct system heterogeneity.

## 5 Understanding Why Pre-training Helps Federated Optimization

While pre-training unsurprisingly speeds up convergence, the reason for the speedup is less apparent. In this section, we examine why pre-training is beneficial to federated learning.

**Pre-training helps align client updates.** To better understand why pre-training alleviates the heterogeneity challenge, we first investigate the gradient diversity of the updates received from different clients. We adopt the notion introduced in Yin et al. [40], adapted here to apply to client

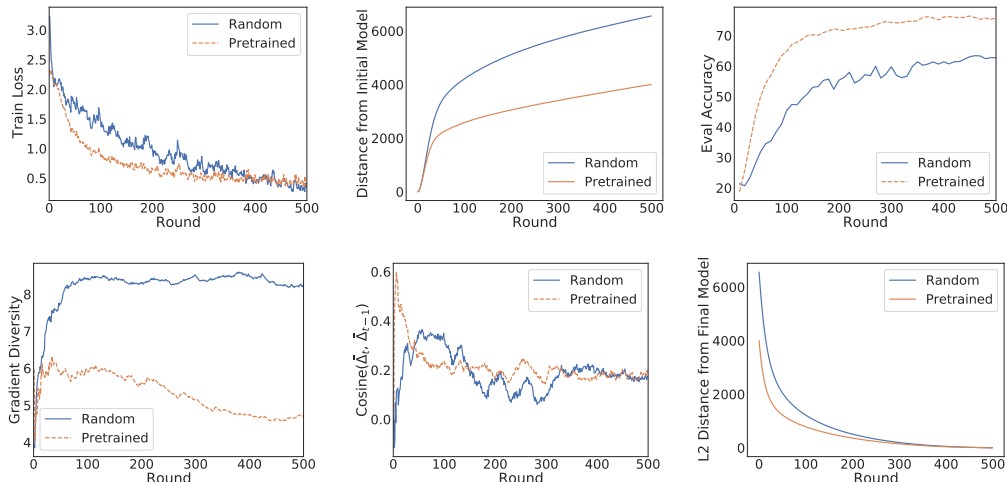

Figure 6: Training and gradient statistics of a Resnet18 on CIFAR-10 with Dirichlet distribution with parameter 0.1. Top row: Train loss of global model; train accuracy of global model; evaluation accuracy of global model; evaluation loss of global model. Bottom row: Gradient diversity of client updates; cosine similarity between client updates; L2 distance of server weights from their final values at the end of training.

|  | CIFAR-10 | FEMNIST | Stack Overflow | Reddit |
|---|---|---|---|---|
| Pre-trained | 661.99 | 26.29 | 151.05 | 647.19 |
| Random | 4843.13 | 355.51 | 185.02 | 1309.68 |

Table 1: The top eigenvalue of the Hessian matrix for each dataset between the pre-trained and random initialized models.

updates $\delta_i$ (whereas Yin et al. [40] focus specifically on gradients $g_i$):

$$\text{GradientDiversity}(\{\Delta_i : i \in \mathcal{S}^t\}) = \frac{\sum_{i \in \mathcal{S}^t} ||\Delta_i||^2}{||\sum_{i \in \mathcal{S}^t} \Delta_i^2||}.$$

In Figure 6, we plot the gradient diversity of client updates $\Delta_i$ at each round for FEDADAM. In the pre-trained setting, client updates have significantly lower gradient diversity (see the bottom left plot in Figure 6). This suggests that when starting from a pre-trained model, the client local model changes are more similar to each other. On the other hand, clients local model changes from randomly initialized weights are almost orthogonal, suffering more from the client drift problem. In addition, when looking at the cosine similarity of consecutive aggregated update vectors in time (bottom middle), we see that consecutive updates point more consistently in a similar direction at the beginning of training when starting from a pre-trained model.

From the top middle and bottom right plots in Figure 6, we see that the pre-trained model starts closer to the final result. We also examine the largest eigenvalue of the Hessian matrix (i.e., local Lipshitz constant or smoothness) at the beginning of training, a larger value of which suggests a harder-to-optimizer loss surface. In Table 1, one can observe that pre-trained models always lead to smaller eigenvalues on different datasets.

**Initial loss for pre-trained versus random models.** Table 2 shows that pre-training does not always lead to lower initial loss. For Squeezenet 1.0 on CIFAR-10 and ResNet-18 on FEMNIST, the initial loss of the randomly initialized models are lower pre-trained models. However, pre-trained model still converges faster as illustrated in Figure 2.

**Connection to theory.** Here, we present the existing optimization theory for FEDAVG and discuss how pre-training helps to improve the model convergence. Following the formulation in Section 3, suppose that there are total $m$ clients, jointly optimizing a global objective function $f(w) = \sum_{i=1}^{m} p_i F_i(w)$,

|  | CIFAR-10 | | FEMNIST | |
|---|---|---|---|---|
|  | Squeezenet 1.0 | ResNet-18 | Squeeze 1.0 | ResNet-18 |
| Pre-trained | 2.71 | 1.07 | 3.99 | 6.58 |
| Random | 1.90 | 1.11 | 4.31 | 4.17 |
|  | Stack Overflow | | Reddit | |
|  | CharLM | DistilGPT2 | CharLM | DistilGPT2 |
| Pre-trained | 6.78 | 4.93 | 5.15 | 6.34 |
| Random | 7.71 | 9.82 | 8.61 | 9.99 |

Table 2: Loss at beginning of training for various model architectures and datasets. The initial loss of the pretrained model is not always lower than that of a random initialization.

and that each client's local loss function $F_i(w)$ is $L$-Lipschitz smooth. For ease of presentation, we assume that all clients participate in training and perform $K$ local SGD updates at each round. Then, under standard assumptions, one can show that after $R$ communication rounds, the expected gradient norm satisfies (see Theorem V in [19]):

$$\mathbb{E}\left\|\nabla f(\bar{x}^R)\right\|^2 \leq \mathcal{O}\left(\frac{\sqrt{F}}{\sqrt{RKm}} + \frac{F^{2/3}\zeta^{2/3}}{R^{2/3}}\right), \tag{2}$$

where $\bar{w}^R$ represents a weighted sampled model from all previous rounds, $\zeta$ is a measure of data heterogeneity, and $F = f(x^0) - f^*$ denotes the gap between the initial loss value and the optimal loss value.

In addition, in order for FEDAVG to achieve the $\mathcal{O}(1/\sqrt{RKm})$ asymptotic convergence rate, previous works [37, 39, 19, 35] showed that the number of local updates $K$ should be upper bounded as follows:

$$K \leq \mathcal{O}\left(\frac{R^{1/3}}{F^{1/3}\zeta^{4/3}m}\right). \tag{3}$$

Clearly, if $F$ becomes smaller starting from a pre-trained model, one can use a larger number of local updates. This corroborates our empirical observations in Figure 3.

When starting from a pre-trained model, the initial gap $F$ is sometimes reduced, as observed in Table 2. As a result, the optimization error upper bound (2) will be smaller, i.e., we get better worst-case performance. However a lower initial loss is not always observed in our experiments, so this does not fully explain our observations, suggesting that we may need to re-think the convergence theory of local update methods.

## 6 Recommendations

In this work, we study the effects of pre-training on federated optimization methods. Our results inform the following recommendations:

1. When evaluating FL algorithms, researchers should experiment with both pre-trained (if available) and random weights, as the initialization can clearly impact the relative performance of different methods, and both initialization schemes may be relevant to practice.

2. When deploying FL to a production environment, using adaptive server optimizers such as FEDADAM together with SGD at the client is a simple and competitive approach when it is possible to start from a pre-trained model.

3. When there is public data to pre-train a model, the impact of heterogeneity can be reduced. Thus, when focusing on heterogeneity, it may be worth considering whether or not proxy data is available for pre-training to motivate the application considered.

## 7 Related Work

**Transfer learning.** Model initialization can significantly impact training and final performance. Previous work studying the loss landscape of deep networks observed significant differences between

the landscape around a random initialization and the landscape later in training. In particular, later in training, the loss can be much more "well-behaved" [23, 10, 9]. Fine-tuning from pre-trained models is common practice in natural language processing and computer vision, yielding strong performance on many tasks [31, 7, 6, 12].

**Federated Optimization.** While a significant amount of research focused on various aspects of FL, including communication-efficiency [26], data and systems heterogeneity [24, 36], and faster convergence rate []. Nearly all previous work in this field neglect the importance of initialization. In our work, we study the impact of initialization on federated optimization in the cross-device setting. We defer the interested reader to surveys of Kairouz et al. [18] and Wang et al. [37] for additional background.

**Pre-training in Federated Learning.** Very few works have studied pre-trained models in federated learning [30, 15, 42, 25, 34]. Zhao et al. [42] studied pre-training as a mechanism to remedy the adverse effect of heterogeneity in FL. However, Zhao et al. [42] found that pre-trained initialization does not alleviate the effect of heterogeneity. In this work, we find that pre-training can alleviate both data and system heterogeneity. Pillutla et al. [30], Hsu et al. [15], Lin et al. [25], Stremmel and Singh [34] experimented with pre-trained models but did not study the difference between random initialization and pre-training, which is the focus of this work. In concurrent and independent works, Chen et al. [4], Weller et al. [38] find that pre-training closes the accuracy gap between FEDAVG and centralized learning under non-IID data. Weller et al. [38] studied multilingual language tasks using large transformer models on synthetically partitioned data, not a realistic setup for large-scale cross-device federated settings. Chen et al. [4] studied the impact of pre-training on data heterogeneity using synthetically partitioned image data. Furthermore, Chen et al. [4] proposed a method to pre-train with synthetic data. In our work, we systematically study both forms of heterogeneity, data-induced and system-induced, across both visual and language tasks on 15 SOTA federated optimization algorithms. We offer theoretical and empirical explanations for why pre-training is beneficial to FL.

## 8   Conclusion and Limitations

**Limitations.** Depending on the application, it may not be possible to get public data, in which case random initialization may be the only option. Nevertheless, we believe there is sufficient prevalence and importance of applications where public data is available for this study to be of broad interest. When public data is available, it may not necessarily reflect the distribution of all users in the population. Consequently, pre-training using public data may introduce bias, which warrants further study, including methods to detect and mitigate such bias. Moreover, we only consider one warm-start initialization strategy: supervised pre-training. Several other possibilities are worth investigating, including meta-learning the warm-start initialization and self-supervised pre-training (e.g., if public data does not come with labels).

**Conclusion.** In this paper, we present a thorough empirical analysis of initialization on federated learning by evaluating it on twelve federated learning algorithms across four vision and text tasks. We find that pre-training on public data can recover most of the accuracy drop from heterogeneity. We show that client updates starting from pre-trained weights have higher cosine similarity, which explains why initializing with pre-trained weights can speed up convergence and achieve high accuracy even in heterogeneous settings. We further show that using simple SGD locally can be as good as other local optimizers.

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

Table 3: Dataset Statistics

| Dataset | Train Clients | Eval Clients | Test Clients | Samples/Client Mean | Std |
|---|---|---|---|---|---|
| CIFAR-10 | 100 | 10 | 10 | 500 | 63 |
| Stack Overflow | 10,815 | 378 | 1,115 | 5,821 | 34,229 |
| FEMNIST | 3,150 | 350 | 350 | 272 | 67 |
| Reddit pushift.io | 9,403 | 4,352 | 4,352 | 34 | 63 |

# A  Experiment Details

## A.1  Datasets and Models

**CIFAR-10.**  We evaluate a multi-class image classification problem on CIFAR-10 [22] using a SqueezeNet [17]. We normalize the images by the dataset mean and standard deviation. Following [14], we partition the dataset using a Dirichlet distribution with parameter 0.1. The statistics on the number of clients and examples in both the training and test splits of the datasets are in Table 3.

**Stack Overflow.**  Stack Overflow consists of questions and answers from Stack Overflow. We experiment a next-word-prediction task using a DistilGPT-2 model with a casual LM head. We perform padding and truncation to ensure that each sentences have 25 words. We then use a GPT-2 tokenizer to encode the tokens.

**FEMNIST.**  Federated EMNIST-62 (FEMNIST) consists of digits and English characters, totaling 62 classes. We evaluate a multi-class image classification problem on the federated version [3] which partitions the digits by the writer and filter out clients that less than one example. As for the model, we use a SqueezeNet 1.0 [17]. Since FEMNIST contains grayscale images, we replicated the one channel value into three channels with the same values.

**Reddit pushift.io.**  Reddit pushift.io contains preprocessed comments posted on the social network on December 2017. The dataset consists of 1,660,820 users totaling 56,587,343 tokens. Due to limited compute capabilities, we sub-sampled 9,403 users for training, 4352 for evaluation and 4352 for test. We evaluate a next-word-prediction task using a CharLM [21] model.

## A.2  Implementation Details

We implemented all algorithms in Pytorch [29] and evaluated them on a cluster of machines, each with eight NVidia V100 GPUs. We evaluate our experiments in FLSim [4]. For all experiments, we tune hyperparameters using Bayesian optimization [33]. We select the best hyperparameters based the final accuracy after a fixed number of rounds for each dataset.

# B  Algorithms Summary

The FEDOPT framework is shown in Algorithm 1.

In this section, we summarize the differences between the SERVEROPT and CLIENTOPT combinations used in our experiments.

## B.1  Hyperparameter Ranges

Below, we show the range for the client learning rate ($\eta_\ell$), server learning rate ($\eta_g$). We fixed server momentum for FEDAVG and FEDADAM $\beta_1$ at 0.9 and proximal term for PROXIMAL to 0.1.

---

[4]https://github.com/facebookresearch/FLSim

Table 4: Comparison of characteristics considered in previous work and the methods analyzed in this paper. Notation: *NA* = Normalized Averaging, *LS* = Non-identical Local Steps, *GM* = Global Momentum, *AS* = Adaptive Server Learning Rate, *GS* = Apply Server Optimizer State Locally.

| | NA | LS | GM | AS | GS |
|---|---|---|---|---|---|
| FEDAVG NOVA | ✓ | ✓ | ✗ | ✗ | ✗ |
| FEDAVG PROXIMAL | ✗ | ✓ | ✗ | ✗ | ✗ |
| FEDAVG SGD | ✗ | ✓ | ✗ | ✗ | ✗ |
| FEDAVG GD | ✗ | ✗ | ✗ | ✗ | ✗ |
| FEDAVG MIMELITE | ✗ | ✓ | ✗ | ✗ | ✓ |
| FEDAVGM NOVA | ✓ | ✓ | ✓ | ✗ | ✗ |
| FEDAVGM PROXIMAL | ✗ | ✓ | ✓ | ✗ | ✗ |
| FEDAVGM SGD | ✗ | ✓ | ✓ | ✗ | ✗ |
| FEDAVGM GD | ✗ | ✗ | ✓ | ✗ | ✗ |
| FEDAVGM MIMELITE | ✗ | ✓ | ✓ | ✗ | ✓ |
| FEDADAM NOVA | ✓ | ✓ | ✓ | ✓ | ✗ |
| FEDADAM PROXIMAL | ✗ | ✓ | ✓ | ✓ | ✗ |
| FEDADAM SGD | ✗ | ✓ | ✓ | ✓ | ✗ |
| FEDADAM GD | ✗ | ✗ | ✓ | ✓ | ✗ |
| FEDADAM MIMELITE | ✗ | ✓ | ✓ | ✓ | ✓ |

$$\beta \in \{0, 0.1, 0.2, 0.3, 0.4, 0.5, 0.6, 0.7, 0.8, 0.9, 0.99\}$$
$$\eta_\ell \in [1 \cdot 10^{-6}, 10]$$
$$\eta_g \in [1 \cdot 10^{-6}, 10]$$

## B.2    Pre-training CharLM

To pre-train CharLM, we train an the CharLM model [21] using a vocab size of 5000. We train the model on Wikitext-103 for 100 epochs using AdamW as the optimizer, learning rate = 0.001, weight-decay = 0.00001, and eps = 1e-8. We will release the code and the pre-train models in the camera-ready version.

## C    Additional Results

### C.1    Fine-tuning only the last layer

In this section, we present the results for fine-tuning only the last linear layer rather in the model as commonly done in practice. Figure 7 shows that fine-tuning only the last layer might not yield optimal model quality and should be consider carefully. While fine-tuning only the last layer can achieve close to full fine-tuning on Stack Overflow, the performance is much worse on CIFAR-10.

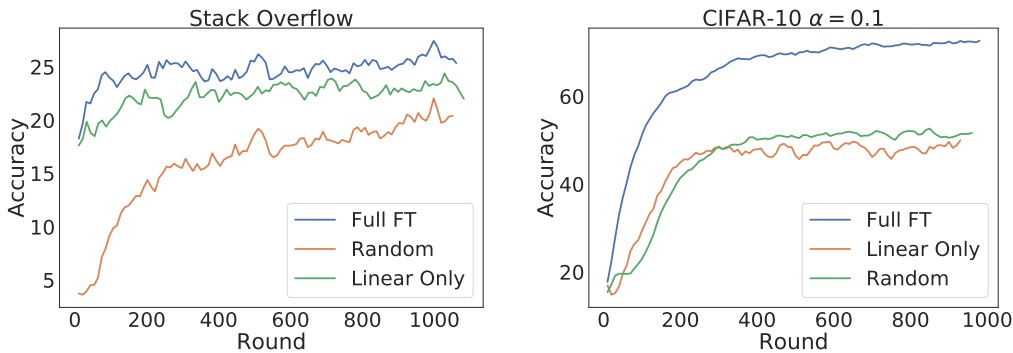

Figure 7: Average accuracy for full fine-tuning, random, and last layer only on Stack Overflow and CIFAR-10.

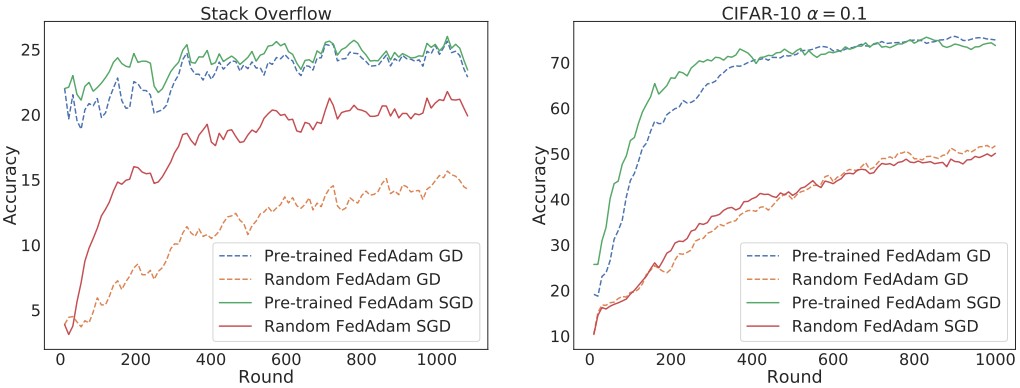

Figure 8: The average accuracy for Stack Overflow and CIFAR-10 comparing FEDADAM with SGD and FEDADAM with full-batch gradient descent (GD).

