# OpenReview forum: "Where to Begin? On the Impact of Pre-Training and Initialization in Federated Learning"
_NeurIPS.cc/2022/Workshop/Federated_Learning — FL-NeurIPS 2022 Poster_

### Official Review · Reviewer_gmNL · 2022-10-14
**Concise empirical study with thorough analysis**

I enjoyed this paper very much, thanks. it is of high quality and a good reference for understanding many empirically relevant aspects of FL.
The only critique I have is that for Figure 1 (and similar ones in the appendix), it took me some time to realize the different y-axes. Maybe a note in the caption would be helpful.

---

### Official Review · Reviewer_hDf4 · 2022-10-17
**The paper examines the impact of model pre-training in federated learning on the performance and training time. The authors compare it to starting from random initialization, using different datasets and optimization techniques from the literature. Their observations demonstrate that pre-training reduces the data and system heterogeneity impact, both of which are known as pivotal challenges in FL.**

Strengths -
1.	The authors study the impacts of pre-training in FL, which although isn’t a new paradigm in FL, but on the other hand, a complete study of its effects on state-of-art benchmarks and frameworks is missing.
2.	Some of the observations are not directly intuitive and raises some questions for future works.
3.	They try to display the impacts of pre-training on both data and system heterogeneity and make some interesting observations.

Weaknesses -
1.	The paper doesn’t seem to have many new observations. The fact that pre-training improves training time is intuitive and is not new work.
2.	While some results on heterogeneity seem attractive, they are not completely studied or explained.
3.	The level of Non-IID-ness is not mentioned and makes the results gained from the observations not convincing.

Details -
1.	According to the observations, there is a small gap between accuracy of pre-trained models on both IID and Non-IID data. Additionally, pre-trained model impact on system heterogeneity are main observations of this paper. However, some of the main points of these observations are missed or not explained.
2.	It is mentioned that Non-IID data is according to natural partitioning without any citation or description about the amount or any metric of Non-IID levels.
3.	Complementary studies with different levels of Non-IID-ness are required to gain the mentioned results about reduction of data heterogeneity from the observations.
4.	There is no explanation or reasoning on the impact of pre-training on system heterogeneity. This is not intuitive at all and it seems that it remains a question for the authors as well.
5.	Figure 5 that seems to be the most informative comparison is located in the Appendix and is not well scaled. It’s also better to use similar colors for the equal methods in different plots of datasets for easier and quicker understanding.
6.	Why does the second plot of figure 5 follow a different approach from other compared plots?

---

### Official Review · Reviewer_fZsu · 2022-10-18
**Paper essentially Investigates Fine-Tuning for Federated Learning**

The paper empirically investigates the impact of model initialization in federated learning (FL) by comparing the performance of several federated optimization algorithms for random initialization and initialization with a pre-trained model.

Strengths:
1. The paper is overall well-written.
2. Experiments are performed on several datasets with a number of federated optimization algorithms.

Weaknesses:
1. The paper claims in Introduction and Sec. 3.2 that “In many FL applications, pre-training can be done on a large non-private dataset available at the server”. It is important to cite references for such applications. The FL setup is motivated by the need to protect privacy of client datasets, coupled with unavailability of non-private datasets. So, it is not clear what are the applications that allow a large non-private dataset at the server.

* Furthermore, if such a large non-private dataset is available at the server, then it does not make sense to discard it and train FL model with random initialization only on clients’ datasets. Sever dataset needs to be used in some manner if it is available, and pre-training seems like a natural way to leverage such a dataset.

2. Experiments in the paper essentially use conventional pre-trained models — ResNet trained on ImageNet and Language Models trained on WikiText. This is equivalent to classical transfer learning, in particular fine-tuning, but in the federated learning setup. The comparison of various federated optimization algorithms for fine-tuning is interesting, however, in my view, it would be more appropriate to position the paper as empirically evaluating federated optimization algorithms for transfer learning (fine-tuning) versus training from scratch, as opposed to studying the impact of model initialization in FL.

* It would be also important to check if there are any prior works that consider fine-tuning in FL, and compare the findings with those works.

While the experiments are interesting, addressing the concerns mentioned in Weaknesses is important. Thus, I think the paper is not yet ready for publication.

Additional comment:

1. Figure 1 is a bit confusing due to interpolated lines. It seems like the plots are for the final test accuracy, and there is no x-axis. Since there is no x-axis, it would be much better to have a table instead of a figure. Currently, the range of left and right axes are often different and a higher value appears at a lower position, making it difficult to interpret the results.

---

### Decision · Program_Chairs · 2022-10-20

Accept (Poster)